# The Pivotal Role of Macrophages in the Pathogenesis of Pancreatic Diseases

**DOI:** 10.3390/ijms25115765

**Published:** 2024-05-25

**Authors:** Seungyeon Ryu, Eun Kyung Lee

**Affiliations:** 1Department of Biochemistry, College of Medicine, The Catholic University of Korea, Seoul 06591, Republic of Korea; s_y_320@catholic.ac.kr; 2Department of Biomedicine & Health Sciences, College of Medicine, The Catholic University of Korea, Seoul 06591, Republic of Korea; 3Institute for Aging and Metabolic Diseases, College of Medicine, The Catholic University of Korea, Seoul 06591, Republic of Korea

**Keywords:** pancreas, macrophage, polarization, pancreatitis, diabetes, pancreatic cancer

## Abstract

The pancreas is an organ with both exocrine and endocrine functions, comprising a highly organized and complex tissue microenvironment composed of diverse cellular and non-cellular components. The impairment of microenvironmental homeostasis, mediated by the dysregulation of cell-to-cell crosstalk, can lead to pancreatic diseases such as pancreatitis, diabetes, and pancreatic cancer. Macrophages, key immune effector cells, can dynamically modulate their polarization status between pro-inflammatory (M1) and anti-inflammatory (M2) modes, critically influencing the homeostasis of the pancreatic microenvironment and thus playing a pivotal role in the pathogenesis of the pancreatic disease. This review aims to summarize current findings and provide detailed mechanistic insights into how alterations mediated by macrophage polarization contribute to the pathogenesis of pancreatic disorders. By analyzing current research comprehensively, this article endeavors to deepen our mechanistic understanding of regulatory molecules that affect macrophage polarity and the intricate crosstalk that regulates pancreatic function within the microenvironment, thereby facilitating the development of innovative therapeutic strategies that target perturbations in the pancreatic microenvironment.

## 1. Introduction

The pancreas is a complex organ that plays a pivotal role in maintaining metabolic homeostasis through its dual physiological functions [1,2]. It operates as an exocrine gland by producing and secreting digestive enzymes, such as amylase, lipase, and trypsin, via the acinar cells, facilitating the breakdown of carbohydrates, fats, and proteins in the digestive tract. Concurrently, it functions as an endocrine gland through the islet cells of Langerhans, which are responsible for the synthesis and release of vital hormones, including insulin, glucagon, and somatostatin, integral to the regulation of blood glucose levels [2,3].

In addition to its primary functions, the pancreatic microenvironment is characterized by a heterogeneous composition of cellular elements, including endothelial cells that constitute the vascular architecture, ductal cells that delineate the pancreatic ducts, neurons responsible for conveying neural signals, and an array of immune cells that regulate immune responses. This elaborate cellular network is supported by an intricate crosstalk of intercellular communication pathways, essential for the coordinated regulation of both exocrine and endocrine activities within the pancreas [4,5,6]. The structural and functional integrity of this environment is critical for the ability of the pancreas to respond to physiological challenges, necessitating precise and harmonious intercellular interactions to maintain the homeostasis of pancreatic function. Abnormal regulations in this crosstalk can lead to a wide spectrum of pancreatic pathologies such as pancreatitis, diabetes, and cancer, highlighting the importance of a comprehensive understanding of the intercellular crosstalk in maintaining pancreatic homeostasis within these microenvironments in health and disease states [7,8,9].

Growing evidence suggests that aberrations in cell-to-cell communications within the pancreatic microenvironment play a critical role in the pathophysiological responses of the pancreas to diverse stimuli [10]. In particular, this review will particularly emphasize the role of immune cells, notably macrophages, in modulating the cellular crosstalk in the pancreatic microenvironment, which, in turn leads, to a disruption of the homeostasis within the pancreatic milieu [11,12,13]. This dysregulation is increasingly recognized as a significant factor in the pathogenesis of several pancreatic diseases, such as pancreatitis, diabetes, and cancer [10,14,15]. This review aims to summarize current findings and offer insights into the mechanistic view of macrophage-mediated alteration and its regulators in the pathogenesis of pancreatic diseases. Through a comprehensive analysis of current research, this review seeks to advance our understanding of the complex interplay between cellular crosstalk and pancreatic function, paving the way for novel therapeutic strategies targeting the microenvironmental disruptions in pancreatic diseases.

## 2. Methodology

This review systematically collected references on macrophages and pancreatic disease from databases, including PubMed and Google Scholar, up to April 2024, using the search terms ‘macrophage’ combined with ‘pancreatitis’, ‘diabetes’, and ‘pancreatic cancer’. After an initial review of the current evidence, we sorted and grouped the articles and reviewed each in its appropriate context. Opinions, letters, and comments were excluded from the analysis.

## 3. Macrophage and Its Polarity

Macrophages, pivotal components of the innate immune system, originate as monocytes within the hematopoietic niche of the bone marrow. These progenitors undergo a well-orchestrated differentiation process driven by a complex interplay of growth factors and cytokines, transitioning into monocytes that circulate transiently within the bloodstream. Upon receiving appropriate chemotactic signals, these monocytes extravasate into peripheral tissues, where they further differentiate into macrophages, adopting phenotypes that are intricately tailored to their specific microenvironmental context [16,17,18]. Macrophages play an important role in the host defense mechanisms, primarily through phagocytosis, wherein they engulf and degrade pathogens, apoptotic cells, and cellular debris. Beyond their phagocytic activity, macrophages are key arbiters of inflammation and tissue repair, secreting an array of cytokines, chemokines, and growth factors that modulate the immune response and orchestrate wound healing and tissue remodeling processes [17,19].

Macrophages exhibit remarkable phenotypic plasticity, enabling them to respond dynamically to a spectrum of immune-metabolic stimuli [20]. This functional versatility is underscored by their capacity to adopt distinct activation states, broadly categorized into the classical (M1) activation state, which is associated with pro-inflammatory, antimicrobial responses, and the alternative (M2) activation state, which is linked to anti-inflammatory, pro-repair activities [17,21]. M2 macrophages have subsets, including M2a, M2b, M2c, and M2d, based on their activation stimuli and unique gene expression profiles [17,22]. The M2a subtype is produced by interleukin-4 (IL-4)/IL-13 stimulation and mediates tissue repair and wound healing. The M2b subtype activated by LPS and IL-1β regulates immune response, while the M2c subtype stimulated by IL-10 exhibits strong anti-inflammatory activity. The M2d subtype resembles features of tumor-associated macrophages (reviewed in [23]).

The nomenclature of macrophages reflects their tissue-specific roles and origins, with Kupffer cells in the liver, microglia in the central nervous system, and osteoclasts in the bone representing well-characterized examples of tissue-resident macrophages [20,24]. Notably, macrophages also populate other tissues, such as the lungs, adipose tissue, and the pancreas, where they contribute to maintaining tissue homeostasis and resolving inflammation [25].

Recent advancements in research have elucidated the multifaceted roles of macrophages and the diverse regulatory elements that influence their function, as illustrated in Figure 1. This body of work highlights the significant impact of macrophages across a wide spectrum of pathophysiological states, encompassing metabolic dysfunctions, fibrotic conditions, oncogenesis, and persistent inflammatory disorders [11,13,19]. The dual nature of macrophages, serving both as guardians in host defense mechanisms and as contributors to the etiology of various diseases, underscores their potential as pivotal targets within therapeutic landscapes [17]. This may provide promising prospects for innovative strategies to fine-tune macrophage activity within disease milieus. This review aims to synthesize contemporary insights into the role and regulatory mechanisms of macrophages in the context of pancreatic pathologies, including pancreatitis, diabetes, and pancreatic cancer.

## 4. Macrophage Activation in Pancreatitis and Its Regulators

Pancreatitis is characterized by the inflammation of the pancreas mediated by the destruction of acinar cells [26,27]. Various factors, including gallstones, alcohol consumption, smoking, hyperglycemia, and genetic mutations, can trigger this pathological state. The disease manifests in two primary forms, distinguished by the duration of inflammation: acute pancreatitis (AP) and chronic pancreatitis (CP) [26,28]. In the acute phase, pancreatic injury leads to the disintegration of acinar cells, which subsequently initiates the activation of immune cells such as macrophages. These activated immune cells enhance the inflammatory response by releasing inflammatory cytokines, such as tumor necrosis factor-alpha (TNF-α), IL-1 beta (IL-1β), IL-6, C-C motif chemokine ligand 2 (CCL2), and exacerbating the condition [29]. Should the inflammatory stimuli persist, atrophic acinar cells engage in the prolonged activation of various immune cells. This sustained immune response culminates in the secretion of various inflammatory cytokines, including IL-1, IL-6, transforming growth factor-beta (TGF-β), and other growth factors, which, in turn, stimulate the activation of pancreatic stellate cells (PSC) [28]. The activation of PSCs leads to excessive extracellular matrix (ECM) production, which is a pivotal event in the progression of chronic pancreatitis, a condition characterized by its potential to induce fibrosis and significantly increase the risk of cancer development within the pancreatic tissue [8,27].

This intricate interplay between cellular destruction, immune activation, and cytokine release underscores the complex pathophysiological mechanisms underlying pancreatitis. Recent studies using several animal models for pancreatitis have shed light on the pivotal role of macrophages in the pathogenesis of pancreatitis. In this section, we aim to explore the characteristics of macrophages identified in the context of pancreatitis, the factors modulating the activation states of macrophages, and their possible roles in the progression of pancreatitis. By delving into the phenotype, activation states, and regulatory mechanisms of macrophages within the pancreatitis milieu, we seek to unravel the nuanced contributions of macrophages to the progression and resolution of the disease.

### 4.1. Factors Regulating Macrophage Polarization in AP

AP is a clinical condition characterized by a sudden inflammation of the pancreas, leading to systemic inflammatory responses and multiorgan dysfunction in severe cases [26,29]. This temporary pancreatic injury disrupts the organ’s ability to function normally, affecting the secretion of digestive enzymes and causing significant metabolic disturbances. Researchers often employ various models to understand the underlying mechanisms and potential therapeutic targets for AP [30]. These include models induced by cerulein (also known as caerulein and ceruletide), which simulate the hyperstimulation of pancreatic acinar cells; sodium taurocholate, which mimics the effects of bile acid reflux into the pancreatic duct; and lipopolysaccharide (LPS), which is used to induce an inflammatory response like that observed in bacterial infections. Each model helps to elucidate different aspects of the disease, from enzyme secretion and acinar cell injury to inflammatory pathways and systemic effects.

Recent studies have particularly emphasized the role of macrophages in AP [27]. Their versatile capacity, exhibiting either pro-inflammatory (M1) or anti-inflammatory (M2) phenotypes, is crucial in initiating and resolving inflammation in AP. The balance between these polarized states of macrophages can significantly influence the progression and severity of pancreatitis, making them key targets for potential interventions [27]. Understanding how various factors and signaling pathways influence macrophage polarization in the context of AP is essential for developing more effective treatments that can modulate the immune response and potentially mitigate the impact of the disease. One of the key regulators involved in this process is the mixed-lineage kinase domain-like protein (MLKL) [31]. It has been found that in cerulein-stimulated acinar cells, a model for AP, there is an upregulation in the expression and phosphorylation of MLKL. Knockout studies have shown that the absence of MLKL leads to decreased levels of CXCL10 in acinar cells and a reduction in M1 macrophage polarization, which subsequently lowers the severity of AP following cerulein and LPS stimulation. MicroRNAs (miRNAs), especially *miR-29a*, also play a significant role in modulating the pathology of pancreatitis [32]. Reports by Dey and colleagues indicate that *miR-29a* is consistently downregulated in murine and human pancreatitis samples. Specifically, the pancreas-specific deletion of the *miR-29a/b1* gene results in a detrimental outcome in response to cerulein administration by enhancing the pancreatic inflammation, immune cell infiltration, and activation of pancreatic stellate cells. Moreover, long non-coding RNA (lncRNA) such as *lncRNA-MM2P* has been shown to have reduced expression in sodium taurocholate-induced models of AP [33]. The overexpression of *lncRNA-MM2P* can decrease M1 macrophage polarization, thereby alleviating pancreatic inflammation.

In addition to these findings, numerous factors that regulate macrophage polarization in various AP models have been reported. Pancreatic phospholipase A2 (PLA2) acts as a pathogenic trigger of AP [34]. Modulating PLA2 activity can reduce M1 macrophage polarization while potentially enhancing M2 polarization, making PLA2 targeting a consistent focus in the development of AP therapies [35]. Zhang and colleagues have developed nanoparticles that include both PLA2 attractants and inhibitors [36]. Applied in a cerulein-induced pancreatitis model, these nanoparticles demonstrated that the inhibition of PLA2 can reduce macrophage polarization, thereby ameliorating the effects of pancreatitis. Dopamine, which possesses anti-inflammatory properties, and its D_2_ receptor can also influence macrophage activation in pancreatitis [37]. The deletion of the D_2_ receptor specifically in myeloid cells has been shown to induce M1 polarization of macrophages, which exacerbates AP [38]. This suggests that dopamine D_2_ receptor pathways could be therapeutic targets to modulate immune responses in pancreatitis. Other agents like growth and differentiation factor 11 (GDF11, also known as BMP-11), tectoridin (Tec), and paeonol (Pae), which are known for their anti-inflammatory functions, can inhibit M1 polarization and, conversely, promote M2 polarization in various study models [39,40,41]. These findings suggest potential therapeutic benefits in targeting these molecules to balance macrophage responses in AP. Moreover, mesenchymal stem cells derived from the placental chorionic plate (CP-MSCs) have been shown to promote the secretion of TNF-stimulated gene 6 (TSG-6), which, in turn, enhances M2 polarization and plays a protective role in the pathology of AP [42].

Conversely, hyperglycemia can exacerbate AP symptoms by activating Notch signaling in macrophages, which promotes M1 polarization [43]. The application of a Notch inhibitor, such as DAPT, has been reported to alleviate symptoms of AP. Additionally, recent studies have shown that the release of extracellular vesicles (EVs) containing *miR-183-5p* increases in acinar cells during cerulein treatment, inducing the M1 polarization of macrophages and subsequently causing acinar cell damage [44]. This emerging evidence underscores the complex interplay of molecular signals that modulate macrophage polarization and their potential implications in AP therapy.

This growing body of evidence underscores the complexity of regulatory mechanisms governing macrophage polarization and its impact on the severity and progression of AP. Continued research into these pathways is crucial for developing targeted therapies that could modulate inflammatory responses in pancreatitis.

### 4.2. Factors Regulating Macrophage Polarization in CP

CP is characterized by irreversible damage to the pancreas, leading to the progressive replacement of both exocrine and endocrine tissues with fibrotic tissue [28,45]. It presents a significant clinical challenge due to its persistent and progressively worsening nature, often culminating in extensive pancreatic fibrosis. Currently, no effective therapies directly target these underlying fibrotic processes, highlighting the critical need for a more profound understanding of the molecular and cellular mechanisms driving this condition [27,45]. The pathogenesis of chronic pancreatitis is complex and remains only partially elucidated, involving a multitude of factors that sustain a chronic inflammatory state and contribute to fibrotic transformations. Importantly, several recent studies have begun to unravel the crucial role of macrophages in developing chronic pancreatitis, suggesting potential new avenues for therapeutic intervention.

Cytokines released from PSCs, notably IL-4 and IL-13, have been demonstrated to increase the population of alternatively activated (M2) macrophages, which, in turn, promote fibrosis [46]. The experimental model of CP in mice, induced by continuous pancreatic injury with cerulein, exhibits the enhancement in leukocyte infiltration, acinar cell loss, and pancreatic fibrosis. This model also shows increased the M2 polarization of macrophages within the pancreas, which is mediated by the release of IL-4 and IL-13 from PSCs. The blockade of IL-4/13 pathways ameliorates CP symptoms, suggesting that releasing these cytokines from pancreatic cells plays a critical regulatory role in the progression of chronic pancreatitis [46]. Beyond acinar cells, T-helper type 2 (Th2) cells and innate lymphoid cells (ILCs) also contribute to the release of IL-4/13 during the pathogenesis of CP [47]. Furthermore, the depletion of regulatory T (Treg) cells promotes a type 2 immune response, thereby enhancing M2 macrophage polarization and exacerbating the deleterious effects of CP insults.

In addition to cytokines, chemokines such as CCL5 play a pivotal role in CP by promoting the infiltration and maintenance of M2 macrophages, which support the fibrotic environment [48]. The influence of post-translational modifications in this process is illustrated by the action of MLN4924, a neddylation inhibitor, which exacerbates pancreatitis by boosting CCL5 secretion through hypoxia-inducible factor 1 alpha (HIF1α)-mediated gene expression. This highlights the critical interaction between cellular stress mechanisms and inflammatory signaling in CP. Additionally, several pharmacological agents, including pirfenidone [49], isoliquiritigenin [50], dasatinib [51], and berberine [52], have been implicated in contributing to the development of CP by enhancing PSC activation and macrophage polarization.

Factors regulating macrophage polarization in pancreatitis are listed in Table 1. Despite the identification of multiple factors that modulate PSC activation and macrophage polarization in the development of CP, the complexity of its pathogenesis presents significant challenges in achieving a full understanding. This underscores the importance of further studies into these regulatory mechanisms to develop targeted therapies that can effectively address the inflammatory and fibrotic processes underlying CP.

## 5. Macrophage Activation in Diabetes and Its Regulators

Diabetes is a systemic disease that manifests due to either a deficiency in insulin production or a failure in insulin action, leading to widespread pathological changes across multiple organ systems. This condition impacts a vast array of bodily functions, disrupting normal metabolic processes and increasing the risk of severe complications, such as cardiovascular disease, kidney damage, and neuropathy [53]. Recent research has highlighted the significant role of macrophages in maintaining the homeostasis of tissue microenvironments within these various organs affected by diabetes and insulin resistance [54,55,56]. These studies suggest that macrophages can influence a range of cellular processes, from inflammatory responses to tissue repair and fibrosis. The intricate interplay between macrophages and islet cells, such as pancreatic β cells and endothelial cells within the islets, as well as the broader interaction between macrophages and peripheral tissues in models of diabetes, are critical for the maintenance of tissue homeostasis, which can exacerbate pathological conditions if dysregulated. This section aims to discuss the characteristics of macrophage polarization in the context of diabetes and insulin resistance and the factors regulating this process.

Islet dysfunction can act as a primary driver of insulin resistance and the onset of diabetes, where the intricate interactions between β cells and other cellular components within the islets are crucial for maintaining islet microenvironment homeostasis [57,58,59]. This intercellular crosstalk in the islet is critical; its disturbances can lead to significant β cell dysfunction. For instance, endothelial cells and macrophages within the islets engage in dynamic communication with β cells, regulating their integrity and function through fine-tuned signaling networks [55,56]. Pancreatic β cells themselves are not passive in this environment; they actively secrete various factors, such as ATP, insulin, and pro-inflammatory molecules, such as colony-stimulating factor 1 (CSF1), islet amyloid polypeptide (IAPP), and IL-1β, which influence the growth, migration, and differentiation of neighboring cells, including macrophages [54,55,60,61,62]. Recent research has shown that pancreatic β cells can secrete factors like Fetuin-A, which increases macrophage infiltration within the islets and promotes intra-islet inflammation [63]. Additionally, advanced glycation end products (AGEs) in the islets have been observed to drive macrophage polarization toward an M1 phenotype, further exacerbating pancreatic β cell dysfunction [64]. Similarly, glucose-regulated protein 94 (GRP94) can promote M1 polarization, thereby enhancing systemic insulin resistance [65]. Interventions such as co-culturing with umbilical cord-derived mesenchymal stem cells (US-MSCs) and glucagon-like peptide-1 (GLP-1) treatment have shown protective effects on β cells against inflammatory responses by decreasing M1 polarization or enhancing M2 polarization of macrophage [66,67]. These studies suggest that macrophage polarization and the regulators that can modulate it within the islets significantly impact β cell dysfunction and could be therapeutic targets in the treatment and management of insulin resistance and diabetes. While research in this field continues to develop, these findings underscore the potential of targeting cellular interactions that contribute to intra-islet inflammation as a strategy to mitigate the progression of insulin resistance and diabetes.

Macrophages play a crucial role in maintaining tissue microenvironment homeostasis within the islets and in various peripheral tissues, including the kidneys and adipose tissues. Understanding the factors that regulate macrophage polarization in peripheral tissues is essential for intervening in insulin resistance, diabetes, and related pathological conditions. Recent studies have demonstrated specific instances where the modulation of macrophage polarization can have therapeutic effects in pathologic conditions related to diabetes. For example, hyperoside (HPS) has been shown to promote M2 polarization in the kidneys of *db*/*db* diabetic mice, improving fasting blood glucose levels and hyperlipidemia, thereby offering protection against diabetic nephropathy [68]. Cell therapy using UC-MSCs has also been applied to diabetic kidneys. Zhang and colleagues reported that *miR-146a-5p* derived from UC-MSCs can enhance M2 polarization and contribute to renal injury recovery [69]. Furthermore, insulin and IL-25 have been found to promote M2 polarization in the diabetic wound environment, enhancing wound healing [70,71].

In models of insulin resistance and diabetes, miRNAs have been reported to regulate macrophage polarization. For instance, *miR-330-5p*, which is induced in high-fat diet (HFD)-induced obesity, promotes M1 polarization exacerbating insulin resistance, while anti-*miR-330-5p* treatment may mitigate insulin resistance [72]. Conversely, the expression of *miR-3061* is reduced in the cecal ligation and puncture (CLP)-induced sepsis models, leading to enhanced M1 polarization and worsening inflammation in diabetic intestines [73]. Furthermore, proline–serine–threonine phosphatase-interacting protein 2 (PSTPIP2) is downregulated in the adipose tissue of diabetic mice. The overexpression of PSTPIP2 decreases M1 polarization and simultaneously enhances M2 polarization, alleviating inflammation in obesity-associated adipose tissue and improving insulin resistance [74].

Macrophage polarization also regulates the diabetic osteoimmune microenvironment [75,76]. Bone morphogenic protein-4 (BMP-4), for instance, increases the M2 polarization of macrophages, thereby accelerating bone defect repair in diabetes mellitus [75]. Additionally, recent studies have reported that specific components from M2 macrophage-derived exosomes can improve diabetic fracture healing [76]. Furthermore, compounds like mogroside V (MV) and 4-octyl itaconate (OI), the cell-permeable itaconate derivate, exhibit anti-inflammatory effects [77,78]. Treatment with MV can mitigate the production of pro-inflammatory cytokines in bone marrow-derived macrophages (BMDM), reducing inflammation [77]. The administration of OI in the mouse models of streptozotocin (STZ)-induced type 1 diabetes (T1D) and spontaneous autoimmune diabetes reduce M1 polarization, alleviating β cell dysfunction and further improving glucose metabolism [78]. Additionally, high-intensity interval training (HIIT) in mice with type 2 diabetes mellitus (T2DM) has been shown to reduce M1 macrophage polarization in the liver, thereby decreasing liver inflammation and improving systemic inflammation associated with T2DM [79]. Factors regulating macrophage polarization in diabetic conditions are listed in Table 2.

## 6. Macrophage Polarization in PDAC and Its Regulators

Pancreatic ductal adenocarcinoma (PDAC) is the most common type of cancer originating in the pancreas and is characterized by a highly complex tumor microenvironment (TME) [80]. This complexity significantly influences the progression of the disease and the response to treatments. The TME of PDAC consists of various cellular and non-cellular components that interact in intricate ways, contributing to the aggressive nature of this cancer. Interactions among components in TME not only support tumor growth and spread but also create a formidable barrier to effective therapeutic interventions, making PDAC notoriously difficult to treat [81,82]. Recent studies have demonstrated that the TME of PDAC is markedly influenced by tissue-associated macrophage (TAM) polarization, particularly the dominance of M2 macrophages [83,84]. Macrophages polarized towards M2, which are pro-tumorigenic and anti-inflammatory, play a crucial role in promoting tumor growth and metastasis by suppressing anti-tumor immune responses and facilitating tissue remodeling and angiogenesis. The abundance of M2 macrophages in the PDAC microenvironment is associated with an immunosuppressive milieu characterized by activated cancer-associated fibroblasts (CAFs), reduced effectiveness of cytotoxic T cells, and enhanced activity of immunosuppressive cells like regulatory T cells (Treg) and myeloid-derived suppressor cells [13,83]. This polarization towards M2 macrophages contributes to chemotherapy resistance and poorer clinical outcomes. Strategies aimed at reprogramming or inhibiting M2 macrophages, thereby altering their recruitment, survival, or function, are being explored as promising approaches to disrupt the supportive tumor niche and enhance treatment efficacy in PDAC [84].

Several proteins, such as insulin-like growth factor binding protein 2 (IGFBP2), transforming growth factor-beta-induced protein (TGFBI), and pyruvate kinase M2 (PKM2), which are upregulated in the pancreatic tissues of PDAC patients, play significant roles in regulating macrophage polarization, thereby influencing PDAC progression [85,86,87]. IGFBP2 is a critical factor in determining TAM polarity in PDAC pathology, where it promotes polarization towards M2 macrophages. This shift increases the infiltration of Tregs and impairs anti-tumor T-cell immunity in mouse models [85]. TGFBI expression is elevated in PDAC, and the downregulation of TGFBI reduces M2 polarization, suppressing macrophage-stimulated tumor growth and enhancing anti-tumor immunity. [86]. Similarly, PKM2 correlates positively with the levels of M2 marker CD206 in PDAC specimens and promotes the M2 polarization of macrophages. The overexpression of PKM2 in PDAC cells enhances the M2 polarization of peripheral blood mononuclear cells (PBMCs) [87]. ALOX5, an enzyme encoded by the *ALOX5* gene known as 5-lipoxygenase, plays a crucial role in M2 macrophage polarization and is a useful druggable target for cancer treatment [88]. A recent study by Hu and colleagues demonstrates that Zileuton, an FDA-approved ALOX5 inhibitor, mitigates the ALOX5-induced invasion and metastasis of PDAC cells [88]. Furthermore, the role of spleen tyrosine kinase (Syk) has been shown to promote PDAC growth and metastasis [89]. The inhibition of Syk using specific chemicals like R788, a Syk inhibitor, exhibits anticancer effects by affecting macrophage polarization [90].

Recent research has highlighted the significant role of extracellular vesicles in the pathology of PDAC [91,92]. Small extracellular vesicles derived from pancreatic cancer cells containing Ezrin (sEV-EZR) are known to potentially facilitate PDAC metastasis by enhancing M2 macrophage polarization [91]. Similarly, exosomes containing the oncogenic KRAS (KRAS^G12D^), which are released through autophagy-dependent ferroptosis, have been observed to promote M2 polarization in TAMs [92]. Additional factors, including Doublecortin-like kinase 1 (DCLK1) isoform 2 [93], a member of the regenerating islet-derived family REG4 [94], NLR family pyrin domain containing 3 (NLRP3) [95], and Notch signaling pathways [96], also play roles in enhancing M2 polarization and have potential roles in cancer progression. Furthermore, non-coding RNAs, including miRNAs and lncRNAs, are critical in regulating the increase in M2 polarization in PDAC. For example, *miR-548t-5p* has a tumor-suppressive role in pancreatic cancer cells [97], and *miR-506* enhances a shift from M2 to M1 polarization, thereby facilitating the anti-tumor immune response in pancreatic cancer [98]. Exosomes that contain lncRNAs *LINC00460* or *FGD5-AS1* contribute to the proliferation and metastasis of pancreatic cancer cells by enhancing M2 polarization in tumor-associated macrophages [99,100]. In addition, innovative strategies like the cytoplasmic delivery of polyinosinic-polycytidylic acid (pIC) and irreversible electroporation (IRE) are being explored to combat pancreatic cancer progression by promoting M1 polarization [101,102]. These approaches suggest potential new directions for therapeutic interventions aimed at modulating macrophage polarization to treat PDAC. Currently identified factors regulating macrophage polarization in PDAC pathology are listed in Table 3.

## 7. Discussion

Several studies have demonstrated the significant role of macrophages in the pathogenesis of pancreatic diseases, highlighting the biological and clinical importance of macrophage polarization in maintaining homeostasis within the pancreatic microenvironment [103,104,105,106]. Since macrophage polarization is a highly dynamic and finely tuned process, further research is needed to fully elucidate the detailed mechanisms underlying this phenomenon. Recent studies have highlighted the role of microbiota in regulating the tissue microenvironment of the pancreas [107,108,109]. Understanding the detailed mechanisms of microbiota-mediated cellular crosstalk in the pancreas is a significant challenge in overcoming pancreatic diseases. In addition to intrinsic factors, extrinsic conditions, such as endoscopic retrograde cholangiopancreatography (ERCP), may also contribute to the induction of pancreatitis [110]. These suggest that the complex and dynamic crosstalk among components of the pancreatic microenvironment is critical for maintaining tissue homeostasis, and its dysregulation is closely related to the pathological process of pancreatic diseases. However, the current research in this area is in the early stages and requires further development. Moreover, the complexity and heterogeneity of macrophage behavior across different stages of pancreatic diseases mean that findings may not be universally applicable. While this review focuses on macrophages, it does not extensively cover other important immune cells and factors in the pancreatic microenvironment. Therefore, to fully understand the mechanisms underlying pancreatic disease, it is essential to pursue a more integrated understanding of intercellular crosstalk within the complex pancreatic microenvironment, composed of diverse components such as islet cells, exocrine cells, acinar cells, immune cells, and microbiota.

Further research is crucial to unravel the complex signaling networks that regulate macrophage behavior. A detailed identification and characterization of the mediators and regulators of macrophage polarization could unveil new therapeutic targets [111,112,113]. Delving deeper into these molecular interactions will enable researchers to develop more effective strategies for preventing, managing, and treating pancreatic diseases. This may include creating novel pharmacological agents specifically aimed at targeting macrophage polarization pathways, thereby altering the disease microenvironment to improve therapeutic outcomes. Additionally, employing advanced technologies such as single-cell RNA sequencing and proteomics can provide exceptional insights into the cellular heterogeneity of the pancreas in both healthy and diseased states [114,115,116]. These technologies are crucial for precisely defining the roles of various macrophage subsets in the progression of pancreatic diseases and may identify previously unknown therapeutic targets.

Tissue-resident macrophages and peritoneal macrophages are related to the pathogenesis of pancreatic diseases such as pancreatitis, cancer, and fibrosis [117,118,119,120]. Tissue-resident macrophages are pivotal in early immune response, inflammation regulation, and tissue repair, maintaining a delicate balance between inflammation and healing. On the other hand, peritoneal macrophages are critical in the acute inflammatory phase, aiding in pathogen clearance and amplifying the immune response. Both types of macrophages are essential for managing inflammation and orchestrating the healing process in pancreatitis, highlighting their importance in disease progression and resolution. Specific markers are typically used to distinguish them: CD68, F4/80, and CD163 for tissue-resident macrophages, and GATA6 or CD11b for peritoneal macrophages. However, it is challenging to clearly define the functions of these two groups of macrophages in pancreatitis due to variations in marker combinations used across different studies and the occasional lack of differentiation between tissue-resident and peritoneal macrophages. Despite these challenges, understanding the roles of these macrophage populations is crucial for elucidating the pathophysiology of pancreatitis and other pancreatic diseases, highlighting the need for more extensive research in this area.

As research advances, it has the potential to yield groundbreaking insights that could transform the treatment landscape for pancreatic diseases. This process would ultimately enhance patient outcomes and pave the way for effective management and potential cures.

## 8. Conclusions

Macrophage polarization within the pancreatic tissue microenvironment plays a crucial role in the pathophysiology of pancreatitis, diabetes, and pancreatic cancer. These conditions reveal that the interactions between macrophages and other cellular components not only affect local tissue responses in the pancreas but also have significant systemic implications. The ability of macrophages to dynamically switch between pro-inflammatory and anti-inflammatory states is key to driving the pathways of inflammation and fibrosis that contribute to the progression of these pancreatic diseases. Understanding the factors that regulate macrophage polarization is essential for uncovering the molecular mechanisms that control systemic processes and cellular interactions within the pancreas. This advancing knowledge enhances our grasp of the complex orchestration of inflammation and immune responses in the pancreatic microenvironment, paving the way for novel therapeutic approaches.

## Figures and Tables

**Figure 1 ijms-25-05765-f001:**
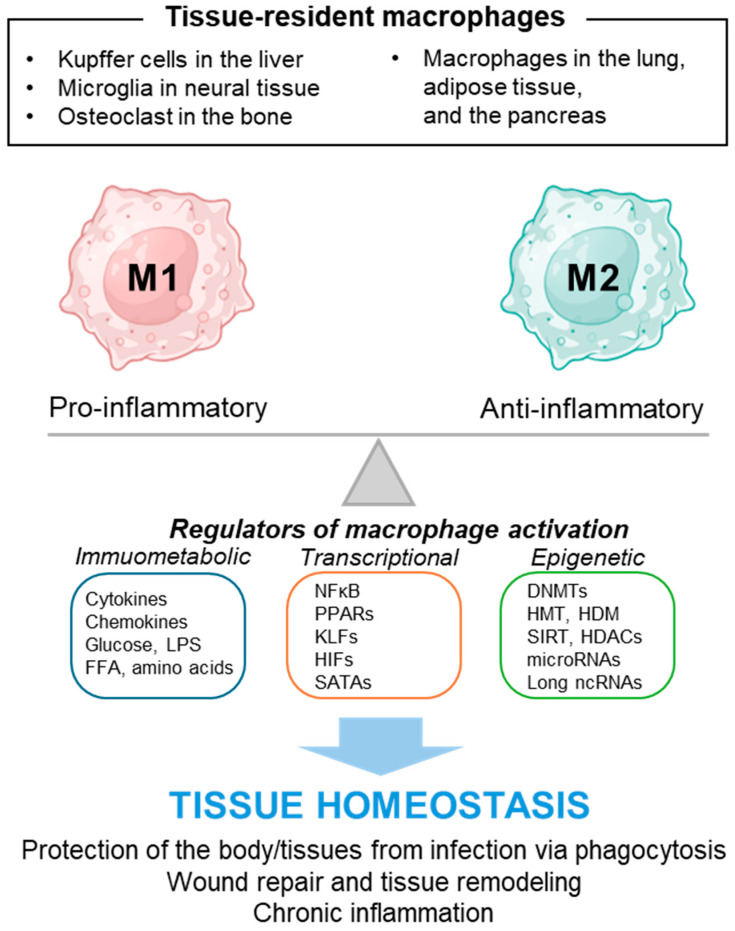
Schematic diagram of macrophage polarization and its regulators. Macrophages respond to changes in the tissue microenvironment by altering their polarity towards M1 or M2, thus playing a role in regulating tissue homeostasis. They engage in phagocytosis to combat infections and contribute to tissue remodeling through wound repair. Additionally, macrophages can induce chronic inflammation, which disrupts the tissue microenvironment. Abnormal macrophage polarization can contribute to the development of pancreatitis, diabetes, and cancer. Various immunometabolic, transcriptional, and epigenetic factors influence macrophage polarization towards M1 or M2 phenotypes, and identifying new regulators is essential for understanding macrophage-mediated tissue homeostasis.

**Table 1 ijms-25-05765-t001:** Factors regulating macrophage polarization in pancreatitis.

Regulators	Mφ Polarization	Effect on Pancreatitis	Ref.
MLKL ↓ in acinar cell	M1 polarization ↓	AP promotion ↓	[31]
*miR-29a/b1* ↓	M1 polarization ↑	AP promotion ↑	[32]
MM2P	M1 polarization ↓	AP promotion ↓	[33]
PLA2 inhibitor	M1 polarization ↓	AP promotion ↓	[36]
Dopamine & D_2_ receptor	M1 polarization ↓	AP promotion ↓	[37,38]
GDF11	M1 polarization ↓	AP promotion ↓	[39]
Tectoridin	M1 polarization ↓	AP promotion ↓	[40]
Paeonol	M1 polarization ↓	AP promotion ↓	[41]
CP-MSCs	M2 polarization ↑	AP promotion ↓	[42]
Hyperglycemia	M1 polarization ↑	AP promotion ↑	[43]
*miR-183-5p* in acinar EVs	M1 polarization ↑	AP promotion ↑	[44]
IL-4/13 in PSCs IL-4/13 in T cells	M2 polarization ↑	CP promotion ↑	[46,47]
MLN4924	M2 polarization ↑	CP promotion ↑	[48]
Pirfenidone	M1 polarization ↓ PSC activation ↓	CP promotion ↓	[49]
Isoliquiritigenin	M1 polarization ↓ PSC activation ↓	CP promotion ↓	[50]
Dasatinib	M1 polarization ↓ PSC activation ↓	CP promotion ↓	[51]
Berberine	M2 polarization ↓ PSC activation ↓	CP promotion ↓	[52]

**Table 2 ijms-25-05765-t002:** Factors regulating macrophage polarization in diabetic conditions.

Tissue/Cell	Regulators	Mφ Condition	Effect on Diabetes	Ref.
Pancreatic islet	Several factors from β cell (insulin, ATP, etc.)	Mφ crosstalk with β cell	β cell integrity Islet inflammation	Reviewed in [54,55]
Pancreatic islet	Fetuin-A from β cell	Mφ accumulation in islet	Intra-islet inflammation	[63]
β cell	AGEs	M1 polarization ↑	β cell dysfunction	[64]
Systemic	GRP94	M1 polarization ↑	Insulin resistance ↑	[65]
β cell	US-MSC	M2 polarization ↑	β cell protection	[66]
β cell	GLP-1	M2 polarization ↑	β cell protection	[67]
Kidney	HPS	M2 polarization ↑	Diabetic nephropathy protection	[68]
Kidney	*miR-146a-5p* from US-MSC	M2 polarization ↑	Renal injury recovery	[69]
Skin	Insulin, IL-25	M2 polarization ↑	Wound healing	[70,71]
Systemic	Anti-*miR-330-5p*	M2 polarization ↑	Insulin resistance ↓	[72]
Intestine in sepsis	*miR-3061* ↓	M1 polarization ↑	Intestinal injury ↑	[73]
Adipose tissue	PSTPIP2	M2 polarization ↑	Inflammation, Insulin resistance ↓	[74]
Bone	BMP-4	M2 polarization ↑	Bone defect repair ↑	[75]
Bone	M2-derived exosome	M2 polarization ↑	Diabetic fracture healing ↑	[76]
Systemic	Mogroside V	M1 polarization ↓	Inflammation ↓	[77]
Systemic	4-Octyl itaconate	M1 polarization ↓	Glucose metabolism ↑	[78]
Liver	HIIT	M1 polarization ↓	Inflammation ↓	[79]

**Table 3 ijms-25-05765-t003:** Factors regulating macrophage polarization in PDAC.

Regulators	Mφ Polarization	Effect on PDAC	Ref.
IGFBP2	M2 polarization ↑	PDAC progression ↑	[85]
TGFBI	M2 polarization ↑	PDAC progression ↑	[86]
PKM2	M2 polarization ↑	PDAC progression ↑	[87]
Zileuton (ALOX5 inhibitor)	M2 polarization ↓	PDAC progression ↓	[88]
R788 (Syk inhibitor)	M2 polarization ↓	PDAC progression ↓	[90]
sEV-EZR	M2 polarization ↑	PDAC metastasis ↑	[91]
Oncogenic KRAS	M2 polarization ↑	PDAC progression ↑	[92]
DCLK1-iso2	M2 polarization ↑	PDAC progression ↑	[93]
REG4	M2 polarization ↑	PDAC progression ↑	[94]
NLRP3	M2a/c/d polarization ↑	PDAC progression ↑	[95]
Notch signaling pathway	M2 polarization ↑	PDAC progression ↑	[96]
*miR-548t-5p*	M2 polarization ↓	PDAC progression ↓	[97]
*miR-506*	M2 polarization ↓	PDAC progression ↓	[98]
Exosomal-*LINK00460*	M2 polarization ↑	PDAC progression ↑	[99]
Exosomal-*FGD5-AS1*	M2 polarization ↑	PDAC progression ↑	[100]
Cytoplasmic delivery of pIC	M1 polarization ↑	PDAC progression ↓	[101]
Irreversible electroporation (IRE)	M1 polarization ↑	PDAC progression ↓	[102]

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
