# Peer review of "The Pivotal Role of Macrophages in the Pathogenesis of Pancreatic Diseases"

_ijms, 2024, doi:10.3390/ijms25115765_

Round 1
Reviewer 1 Report
Comments and Suggestions for Authors
The review "The pivotal role of macrophages in the pathogenesis of pancreatic diseases" brings new insights from the literature and organizes them effectively to summarize the implications of macrophages and more in pancreatic diseases. Here are some recommendations:
1. Add a materials and methods chapter discussing how you conducted the literature research.
2. The future perspectives chapter should be separate, and additionally, include a few lines about the study's limitations. Regarding future perspectives, add information about the importance of fecal microbiota transplantation in pancreatic inflammation and pancreatic pathology in general: https://doi.org/10.3390/diagnostics14090861.
3. Add a discussion chapter presenting other articles from the literature with similar themes. Furthermore, discuss the importance of inflammation and pancreatitis-like reactions in patients undergoing ERCP - a highly utilized method today: https://doi.org/10.3390/jpm13091356.
4. The conclusions chapter should be shortened and as concise as possible, without citations, but merely summarizing the information from the review.
Comments on the Quality of English LanguageMinor editing of English language required.
Author Response
- Add a materials and methods chapter discussing how you conducted the literature research.
-->We revised the text as the Reviewer advised.
- The future perspectives chapter should be separate, and additionally, include a few lines about the study's limitations. Regarding future perspectives, add information about the importance of fecal microbiota transplantation in pancreatic inflammation and pancreatic pathology in general: https://doi.org/10.3390/diagnostics14090861.
--> We are grateful for the Reviewer’s critical insight. We believe the role of microbiota is very important to the microenvironment in the pancreas, as suggested by the Reviewer and in literature and others. In response, we cited the reference and revised the text as the Reviewer advised.
- Add a discussion chapter presenting other articles from the literature with similar themes. Furthermore, discuss the importance of inflammation and pancreatitis-like reactions in patients undergoing ERCP - a highly utilized method today: https://doi.org/10.3390/jpm13091356.
--> We are thankful for the Reviewer’s comments. We revised the text by including additional references and adding the discussion section. Although the sections on methodology and discussion are not mandatory based on the editorial office's advice and the instructions for authors provided by the IJMS journal, we revised the text to respond to the Reviewer's comment. Even if our revised manuscript did not discuss specific topics in depth, we hope the reviewer will understand and be generous with this matter.
- The conclusions chapter should be shortened and as concise as possible, without citations, but merely summarizing the information from the review.
--> We are grateful to the Reviewer. We revised the text as the Reviewer advised.
Minor editing of English language required.
--> We revised the text as the Reviewer advised.
Reviewer 2 Report
Comments and Suggestions for Authors
In the manuscript (ID:ijms-3006062), Seungyeon Ryu and Eun Kyung Lee summarized the latest findings on the role of macrophages in the pathogenesis of pancreatic diseases, which makes it an interesting topic. Overall, this is a well-written paper. However, there are a few suggestions and comments. This will help readers understand better. I recommend considering major and minor revision for publication.
Major points
1. M2 macrophages may be further subdivided into four subtypes: M2a, M2b, M2c, M2d.
The authors should provide more detailed information about M2 macrophage.
2. Tissue-resident macrophages and peritoneal macrophages are related in pancreatitis.
The addition of information about tissue-resident macrophages and peritoneal macrophages would increase the significance.
Minor point
1. The authors used three colors in Table 1. What do the different colors mean?
Author Response
Major points
- M2 macrophages may be further subdivided into four subtypes: M2a, M2b, M2c, M2d.
The authors should provide more detailed information about M2 macrophage.
--> We are thankful for the Reviewer’s comments. We revised the text as the Reviewer advised.
- Tissue-resident macrophages and peritoneal macrophages are related in pancreatitis.
The addition of information about tissue-resident macrophages and peritoneal macrophages would increase the significance.
-->We are grateful for the Reviewer’s valuable insight. We expanded our discussion in the revised manuscript.
Minor point
- The authors used three colors in Table 1. What do the different colors mean?
--> We are thankful for the Reviewer’s comments. We revised Table 1.
Reviewer 3 Report
Comments and Suggestions for Authors
In the review "The pivotal role of macrophages in the pathogenesis of pancreatic diseases" the author summarize the current findings and explains the mechanistic view of macrophage-mediated alteration and its regulators in the pathogenesis of pancreatic diseases. In this review the author is analyzing the complex interplay between cellular crosstalk and pancreatic function, searching for novel therapeutic strategies targeting the disruptions in the pancreatic microenvironment.
This review is well written and gives a good overview over pancreatic diseases and the role of macrophages during disease progression and therapeutic strategies.
Author Response
We appreciate the Reviewer’s positive evaluation.
Round 2
Reviewer 1 Report
Comments and Suggestions for Authors
The authors have successfully made the recommended changes.
Reviewer 2 Report
Comments and Suggestions for Authors
The authors have responded appropriately to my concerns.
I recommend this manuscript for publication in IJMS.